# Omega-3 Polyunsaturated Fatty Acids and Stroke Burden

**DOI:** 10.3390/ijms20225549

**Published:** 2019-11-07

**Authors:** Yuji Ueno, Nobukazu Miyamoto, Kazuo Yamashiro, Ryota Tanaka, Nobutaka Hattori

**Affiliations:** 1Department of Neurology, Juntendo University Faculty of Medicine, Tokyo 113-8421, Japan; nobu-m@juntendo.ac.jp (N.M.); kazuo-y@juntendo.ac.jp (K.Y.); nhattori@juntendo.ac.jp (N.H.); 2Stroke Center and Division of Neurology, Department of Medicine, Jichi Medical University, Tochigi 329-0498, Japan; r_tanaka@juntendo.ac.jp

**Keywords:** stroke, omega-3 PUFA, eicosapentaenoic acid, docosahexaenoic acid

## Abstract

Stroke is a major leading cause of death and disability worldwide. N-3 polyunsaturated fatty acids (PUFAs) including eicosapentaenoic acid and docosahexaenoic acid have potent anti-inflammatory effects, reduce platelet aggregation, and regress atherosclerotic plaques. Since the discovery that the Greenland Eskimo population, whose diet is high in marine n-3 PUFAs, have a lower incidence of coronary heart disease than Western populations, numerous epidemiological studies to explore the associations of dietary intakes of fish and n-3 PUFAs with cardiovascular diseases, and large-scale clinical trials to identify the benefits of treatment with n-3 PUFAs have been conducted. In most of these studies the incidence and mortality of stroke were also evaluated mainly as secondary endpoints. Thus, a systematic literature review regarding the association of dietary intake of n-3 PUFAs with stroke in the epidemiological studies and the treatment effects of n-3 PUFAs in the clinical trials was conducted. Moreover, recent experimental studies were also reviewed to explore the molecular mechanisms of the neuroprotective effects of n-3 PUFAs after stroke.

## 1. Introduction

There were an estimated 6.33 million deaths due to stroke and 42.43 million patients suffering from stroke according to the Global Burden of Diseases, Injuries, and Risk Factors 2015 study [1]. In particular, stroke burden and mortality were greater in low-income countries [1,2]. Although age-standardized rates of stroke mortality recently decreased by 30% worldwide, the absolute numbers of stroke patients are increasing because of a reduction in stroke case fatality rates, as well as population growth and aging [3].

Hypertension, diabetes mellitus, dyslipidemia, and atrial fibrillation are fundamental risk factors for stroke [4,5,6]. Emerging data have indicated that lifestyle risk factors such as smoking, high body mass index, low physical activity, and high alcohol consumption could be important risk factors for the development of stroke [7,8,9]. In a population-based cohort study of 31,696 Swedish women, decreasing the number of these lifestyle risk factors was associated with reduction of stroke, especially cerebral infarction [8]. It was shown that stroke survivors who continued to smoke had a two-fold higher risk for recurrent stroke, while physical inactivity increased mortality and the incidence of stroke by 25% to 30% [10,11,12]. Thus, management of not only fundamental atherosclerotic vascular risk factors, but also lifestyle risk factors is critical in stroke prevention. More importantly, of these lifestyle risks, dietary habits were shown to be the most relevant to stroke [12,13,14,15,16]. Among the sorts of foods, there is evidence that every 200 g/day in intake of fruits and vegetables is correlated with a decrease in the risk of stroke by 32% and 11%, respectively [17]. Little information has been available on the association of stroke with ketogenic (low carbohydrate–high fat) and vegetarian and vegan (plant-based) diets, while the Atkins (low carbohydrate–high protein) diet could increase the stroke risk [18].

The n-3 polyunsaturated fatty acids (PUFAs) are a class of essential fatty acids required for normal biological activity and function in living organisms. The n-3 PUFAs are poorly synthesized in the human body, and they are composed of marine fish-derived agents such as eicosapentaenoic acid (EPA) and docosahexaenoic acid (DHA). It is well known that n-3 PUFAs reduce the risk of cardiovascular diseases through multiple actions, including an anti-inflammatory effect, reduction of platelet aggregation, and stabilization of atherosclerotic plaques [19,20,21,22,23,24]. Thus, less dietary intake of n-3 PUFAs can result in an increase in the incidence of cardiovascular diseases.

Considerable insight has been gained into the n-3 PUFAs, through their dietary intake and their use as therapy for the risk of cardiovascular diseases. However, the association of ingestion of n-3 PUFAs with stroke burden has not been adequately studied. The current article reviews the associations of dietary n-3 PUFA intake with stroke and the effects of treatment with n-3 PUFAs on stroke in the previous high-impact studies, and moreover, the mechanistic role of n-3 PUFAs and their metabolites related to stroke burden was investigated.

## 2. The Impact of Dietary Intake of n-3 PUFAs on Stroke

It is well known that dietary habits, especially fish consumption, differ by race, religion, country, and continent. So far, many large-scale population studies to determine the relationships of n-3 PUFAs consumed as food with disease have been carried out in the USA, Europe, and Eastern Asia. In this section, the links of dietary fish intake with stroke incidence and mortality in association with the amount of fish consumed among the different continents are reviewed.

### 2.1. United States of America

Morris et al. first studied the dietary intake of fish to determine the risk of cardiovascular diseases and stroke in 21,885 men. They classified subjects into consuming <1, 1, 2–4, and ≥5 fish meals/week, but a higher frequency of fish consumption was not shown to lower the incidence of fatal and non-fatal stroke [25]. Another prospective cohort study of 79,839 middle-aged women without a history of cancer, cardiovascular diseases, diabetes, or hypercholesterolemia explored the incidence of stroke and classified the subjects by their frequency of fish consumption into ≥5, 2 to 4, and 1 times per week, 1 to 3 times per month, and <1 time per month. For total stroke, compared to women who took fish less than once per month, there was a relative trend for reduction of total stroke in women who ate fish 1–3 times per month (relative risk (RR), 0.93; 95% confidence interval (CI), 0.65–1.34), 1 per week (RR, 0.78; 95% CI, 0.55–1.12), 2–4 per week (RR, 0.73; 95% CI, 0.47–1.14), and ≥5 per week (RR, 0.48; 95% CI, 0.21–1.06). Especially in lacunar stroke, the frequency of consumption of fish (≥2 per week, RR, 0.28; 95% CI, 0.12–0.67) and the highest quintile of intake of n-3 PUFAs (RR, 0.37; 95% CI, 0.19–0.73) were associated with lower risks of stroke development. Moreover, a higher frequency of fish consumption was not related to the incidence of hemorrhagic stroke [26]. In 43,671 men, although there were no significant associations of fatal and non-fatal stroke with the above categorical classification of fish-eating frequency, after dichotomization, fish consumption of at least once per month was significantly related to a lower incidence of ischemic stroke (RR, 0.56; 95% CI, 0.38–0.83), but not hemorrhagic stroke (RR, 1.36; 95% CI, 0.48–3.82), compared with those who ate fish less than once per month [27].

### 2.2. Europe

In the Netherlands, a study of 20,069 men and women showed that the highest quartile of EPA/DHA intake (median 225 mg/d) and fish consumption (median 18 g/d) was linked with a lower stroke incidence compared to the lowest quartile of EPA/DHA intake (median 36 mg/d) and fish consumption (median 1 g/d) in women (hazard ratio (HR), 0.49; 95% CI, 0.27–0.91). However, there was no significant difference in men [28]. The study in Sweden showed that the highest quintile of intake of long chain n-3 PUFAs (median 559 mg/d) was associated with a low incidence of total stroke compared with the lowest quintile (median 131 mg/d) (RR, 0.84; 95% CI, 0.72–0.99) in women, while the other Swedish study did not show any association of ischemic stroke reduction with the amount of n-3 PUFA intake [29,30]. Recent data from a large-scale cohort study in Denmark, which involved 57,053 participants with 13.5 years of follow-up, explored the significance of the association between the incidence of ischemic stroke and n-3 PUFA intake. The results showed that the highest quartiles of total n-3 PUFA intake (RR, 0.69; 95% CI, 0.50–0.95) and EPA intake (HR, 0.66; 95% CI, 0.48–0.91) were associated with the incidence of large artery atherosclerosis among ischemic stroke subtypes [31].

### 2.3. Eastern Asia

A study in Hiroshima and Nagasaki, Japan, demonstrated that fish consumption ≥46 g/day was associated with lower mortality from total stroke by 15% compared with <18 g/day (HR, 0.85; 95% CI, 0.75–0.98) [32]. In particular, a significant reduction in mortality from intracranial hemorrhage (HR, 0.70; 95% CI, 0.54–0.92), but not ischemic stroke (HR, 0.94; 95% CI, 0.77–1.14) was found [32]. The National Integrated Project for Prospective Observation of Non-communicable Diseases and Its Trends in the Aged, 1980 (NIPPON DATA80) encompassed 3945 men and 4934 women, and it showed that fish intake twice a day did not show any beneficial associations with all-cause death (HR, 0.99; 95% CI, 0.77–1.27), stroke death (HR, 1.26; 95% CI, 0.70–2.29), death from cerebral infarction (HR, 1.09; 95% CI, 0.48–2.43), and death from cerebral hemorrhage (HR, 0.92; 95% CI, 0.20–4.23) compared to subjects who ate fish 1 to 2 times weekly [33]. The Japan Collaborative Cohort Study for Evaluation of Cancer Risk study enrolled 57,972 Japanese participants and showed that the highest quintile of fish consumption (median intake, 85 g/day) was associated with a lower risk of death from cardiovascular diseases (RR, 0.82; 95% CI, 0.71–0.95) than the lowest quintile (median intake, 20 g/day). The highest quintile of fish consumption was not related to a low risk of death from total stroke (RR, 0.91; 95% CI, 0.74–1.13), intraparenchymal hemorrhage (RR, 0.95; 95% CI, 0.62–1.47), subarachnoid hemorrhage (RR, 0.96; 95% CI, 0.55–1.68), and ischemic stroke (RR, 0.93; 95% CI, 0.65–1.34) [34]. A Chinese study of 18,244 men showed that death due to acute myocardial infarction (AMI) was infrequent in men eating more than 200 g/week of fish/shellfish compared to men eating less than 50 g/week (RR, 0.41; 95% CI, 0.22–0.78), as well as in those consuming more than 1.10 g/week of n-3 PUFAs compared to less than 0.27 g/week (RR, 0.43; 95% CI, 0.23–0.81). However, there were no associations of stroke mortality with the amount of consumption of fish/shellfish (RR, 1.11; 95% CI, 0.83–1.47) and n-3 PUFAs (RR, 1.00; 95% CI, 0.75–1.33) [35]. Another study of two prospective cohort studies of 134,296 Chinese men and women explored the mortality from cardiovascular diseases with median fish intake for the first to fifth quintiles of 10.8, 25.0, 39.1, 59.8, and 107.2 g/day, respectively, for men, and 10.4, 24.3, 38.5, 58.7, and 105.2 g/day, respectively, for women. Mortality from ischemic stroke was infrequent in men and women with fifth quintile intake (HR, 0.63; 95% CI, 0.41–0.94) compared to those with first quintile intake. Subjects with fifth quintile intake of 0.07 g/day of EPA and 0.15 g of DHA showed lower mortality due to ischemic stroke (HR, 0.63; 95% CI, 0.41–0.94) than subjects with first quintile intake of 0.006 g/day of EPA and 0.009 g of DHA [36]. In the Singapore Chinese Health Study with 63,257 subjects, consumption of EPA and DHA showed a trend towards a lower risk for stroke mortality, but no significant interaction was found, whereas EPA and DHA intakes were inversely associated with the risk of cardiovascular deaths from ischemic heart diseases (HR, 0.86; 95% CI, 0.74–0.99) [37].

### 2.4. Review of the Epidemiological Studies Worldwide and Their Limitations

The association of dietary intake of fish and n-3 PUFAs with stroke burden has been studied worldwide, suggesting that higher intake of fish and n-3 PUFAs might be associated with lower stroke incidence and mortality. A meta-analysis has been carried out, which indicated that dietary fish consumption was related to a lower incidence of stroke, especially in women and those with BMI <24 kg/m^2^, and fatal stroke [38]. Of the sorts of fish, it was shown that fatty fish contains higher amount of n-3 PUFAs than lean fish, but a recent meta-analysis demonstrated that both fatty and lean fish intakes were associated with a reduction of stroke [39].

However, it must be noted that dietary habits differ by race, religion, area, and country when interpreting the data of these studies. For instance, median intakes of n-3 PUFAs in the lowest and highest groups of the enrolled subjects in the studies conducted in the USA, China, Sweden, and Japan were 0.077 and 0.481, 0.59 and 1.26, 0.7 and 1.4, and 1.0 and 2.3 g/day, respectively [26,29,37,40]. Thus, dietary intakes of fish were higher in the order of Japan, Sweden, China, and the USA, which was consistent with the data regarding the global survey of n-3 PUFA levels in healthy adults [41]. The amount of fish consumption in the Japanese subjects generally seemed to be above the threshold in studies in other countries or continents, which might be associated with lower incidences of cardiovascular diseases in Japan compared to Western countries [5].

A recent study indicated that memory-based dietary assessment that was used in these epidemiological studies critically lacked established scientific facts and analytic truths, in terms of the false assumption that human memory of past ingestive behavior was accurate, the requirement to submit inappropriate protocols such as procedures inducing false recall, and data resources derived from subjective mental phenomena [42]. Additionally, some of the epidemiological studies also investigated the consumption of other foods, indicating that higher fish intake was correlated with higher intakes of meat, poultry, vegetables, and fruits [26,27,34,36]. Thus, one cannot exclude the possibility of other food affecting stroke incidence. Collectively, dietary fish intake can be beneficial for the prevention of stroke, but insufficient data have been available.

## 3. Therapeutic Use of n-PUFAs and Stroke Burden

### 3.1. Systematic Literature Review

Around the same time as the epidemiological studies to investigate the efficacy of dietary fish intake for the risk of cardiovascular diseases and stroke were starting to be published, clinical trials to determine whether treatment with n-3 PUFAs lowers the incidences of stroke and cardiovascular diseases had begun. A systematic literature review was performed to search for studies investigating the association of stroke with n-3 PUFA treatment according to the PRISMA guidelines [43]. The search was restricted to articles written in English and performed using PubMed and Medline by entering the search terms “n-3 polyunsaturated fatty acid”, “eicosapentaenoic acid”, or “docosahexaenoic acid”, “cardiovascular”, “randomized”, and “treatment”. Stroke was not used because no studies set stroke as a primary endpoint. Studies were included if they met each of the following criteria: Directly relevant with n-3 PUFA supplementation compared to a placebo supplement; primary outcome measurements including cardiovascular diseases, deaths, or stroke; stroke independently set as a secondary endpoint; the published data available in full text; and only human, randomized, controlled clinical trials. In the studies showing sub-analysis data from large-scale clinical trials, the studies with main results including stroke as an outcome were searched.

Of the 919 publications, 886 studies were identified on 20 August 2019. Two independent investigators (N.M. and K.Y.) conducted the literature search and selection of articles. Potential discrepancies were resolved by open discussion. Details of the search and article selection are summarized in the flow diagram (Figure 1). Studies were included if they were published as original articles investigating the prognosis in subjects receiving n-3 PUFA therapy. Review articles, meta-analyses, case reports, editorial comments, letters, meeting abstracts, and studies not fulfilling our inclusion criteria for their content were excluded.

Some potential limitations of the current study must be considered when interpreting the systematic literature review. First, 8 articles were included in the systematic review, in which the primary endpoints were death, cardiovascular death, and cardiovascular incidents. In all studies, stroke was not set as a primary endpoint, but was one of the secondary endpoints. Second, the enrolled subjects were diverse, such as having atherosclerotic vascular risk factors without cardiovascular diseases, cardiovascular diseases, and chronic heart failure, and no studies specifically focused on the effect of n-3 PUFAs in stroke patients. Thus, there could be a potential bias in the selection of enrolled subjects. Third, trial protocols and sample sizes were different among studies.

### 3.2. EPA/DHA

In Italy, the Gruppo Italiano per lo Studio della Sopravvivenza nell’Infarto miocardico (GISSI)-Prevenzione trial enrolled 11,324 survivors of a recent MI within 3 months of onset and randomly assigned patients to n-3 PUFAs (1 g of EPA and DHA daily in an average ratio of EPA/DHA 1:2), vitamin E (300 mg daily), both, or placebo. Treatment with n-3 PUFAs, but not with vitamin E, decreased primary endpoints including death, nonfatal MI, and nonfatal stroke (RR: 0.90, 95% CI: 0.82%–0.99%, by two-way analysis; RR: 0.85, 95% CI: 0.74%–0.98%, by four-way analysis) [44]. However, fatal and non-fatal strokes were not significantly reduced after n-3 PUFA therapy (RR: 1.21, 95% CI: 0.91%–1.63%, by two-way analysis; RR: 1.30, 95% CI: 0.87%–1.96 %, by four-way analysis) [44]. In 2008, the Gruppo Italiano per lo Studio della Sopravvivenza nell’Infarto miocardico-heart failure (GISSI-HF) studied patients with chronic heart failure of New York Heart Association class II–IV and randomly assigned them to receive 1 g n-3 PUFAs per day (850–882 mg EPA and DHA at an average ratio of 1:1.2) or to matching placebo, as well as rosuvastatin 10 mg daily or placebo, and the primary endpoints included time to death and time to death or admission to hospital due to cardiovascular diseases. Treatment with n-3 PUFAs significantly reduced deaths (HR: 0.91; 95.5% CI: 0.833–0.998) and the occurrence of cardiovascular diseases (HR: 0.92; 99% CI: 0.849–0.999), whereas 10 mg of rosuvastatin did not reduce the occurrence of the primary endpoints [45,46]. The proportions of stroke, ischemic stroke, hemorrhagic stroke, and unknown stroke in the n-3 PUFA treatment and placebo groups were 2.8% and 2.3%, 0.4% and 0.3%, and 0.3% and 0.4%, respectively. Overall stroke incidence during follow-up was slightly higher in the n-3 PUFA treatment group than in the placebo group (3.5% vs. 3.0%, HR: 1.16, 95% CI: 0.89–1.51) [45,46]. The Supplémentation en Folates et Omega-3 (SU.FOL.OM3) trial included 2501 patients with a history of MI, unstable angina, and ischemic stroke and randomly assigned them to n-3 PUFAs (0.6 g of EPA and DHA in the ratio of EPA/DHA 2:1), a dietary supplement containing 5-methyltetrahydrofolate (560 μg), vitamin B-6 (3 mg), and B-12 (20 μg), both, or placebo, in a 2 × 2 factorial design. Treatment with n-3 PUFAs did not reduce non-fatal MI, ischemic stroke, and death from cardiovascular disease (HR: 1.08, 95% CI: 0.79–1.47%), as well as fatal and non-fatal stroke (HR: 1.04, 95% CI: 0.62 vs 1.75%) [47]. The Outcome Reduction with an Initial Glargine Intervention (ORIGIN), a Study of Cardiovascular Events in Diabetes (ASCEND), and the Vitamin D and Omega-3 Trial (VITAL) trials enrolled 25,871 patients at high risk for cardiovascular diseases and diabetes or prediabetes, 15,480 patients with diabetes but without evidence of atherosclerotic cardiovascular disease, and 25,871 men ≥50 years and women ≥ 55 years of age, respectively. In these studies, 1-g capsules of 840 mg of marine n-3 fatty acids (460 vs 465 mg of EPA and 375–380 mg of DHA) did not significantly reduce the incidence of not only cardiovascular diseases and deaths, but also stroke [48,49,50] (Table 1).

### 3.3. EPA

In the Japan EPA Lipid Intervention Study (JELIS), with a prospective, randomized, open-label, and blinded-endpoint (PROBE) design involving 18,645 patients (primary prevention: *n* = 14,981, and secondary prevention: *n* = 3664 for coronary artery diseases), the efficacies of EPA 1800 mg combined with low-dose statins and low-dose statins alone were compared. The primary endpoint was a major coronary event, such as sudden cardiac death, fatal and non-fatal MI, and other non-fatal coronary events including unstable angina pectoris, angioplasty, stenting, or coronary artery bypass grafting. In this study, all patients received 10 mg pravastatin or 5 mg simvastatin as baseline statin therapy. During the follow-up of 4.6 years, the primary endpoint occurred in 262 (2.8%) patients in the EPA with low-dose statins group and in 324 (3.5%) patients in the statins alone group [51]. In a sub-analysis of JELIS, the effect of EPA on stroke incidence was investigated according to the presence of a history of stroke: Primary prevention of stroke, *n* = 17,703; and secondary prevention of stroke, *n* = 942. During follow-up, for the primary prevention of stroke, the rates of ischemic stroke, hemorrhagic stroke, and unknown stroke in the EPA with low-dose statins and in the statins alone groups were 1.0% and 0.9%, 0.5% and 0.3%, and 0.0% and 0.0%, respectively. For the secondary prevention of stroke, the rates of ischemic stroke, hemorrhagic stroke, and unknown stroke in the EPA with low-dose statins and in the statins alone groups were 5.8% and 8.5%, 1.0% and 2.0%, and 0.0% and 0.2%, respectively. Thus, in the secondary prevention groups, EPA significantly suppressed stroke incidence (HR, 0.80; 95% CI, 0.64 to 0.997) in the low-dose statins group, while there was no beneficial effect of EPA combined with statin therapy in the primary prevention group (HR, 1.08; 95% CI, 0.95 to 1.22) [52]. In the Reduction of Cardiovascular Events with Icosapent Ethyl–Intervention (REDUCE-IT) trial, 8,179 patients with established cardiovascular disease or with atherosclerotic vascular risk factors, who had been treated with statins and had an LDL-C level of 41 to 100 mg/dL and a fasting triglyceride level of 150 to 499 mg/dL, were enrolled [53]. REDUCE-IT showed that 4 g of EPA decreased cardiovascular death, nonfatal MI, nonfatal stroke, coronary revascularization, or unstable angina (HR, 0.75; 95% CI, 0.68–0.83) [54]. Moreover, fatal and non-fatal stroke, one of the secondary composite endpoints, was decreased by 4 g of EPA therapy (HR, 0.72; 95% CI, 0.55–0.93) [54].

### 3.4. Insights into Controversial Results from Clinical Trials

As mentioned above, many large-scale clinical studies have been carried out, but most of them did not show any efficacy of n-3 PUFA therapy to reduce stroke and cardiovascular diseases. As a possible explanation, the dose of n-3 PUFAs might have been insufficient in GISSI-P, GISSI-HF, ORIGIN, ASEND, and VITAL (<1 g, daily) [44,45,46,48,49,50]. On the other hand, 1800 mg of EPA showed a significant reduction in stroke incidence in the JELIS study in the secondary stroke prevention subgroup, and moreover, 4 g of EPA reduced stroke incidence in the REDUCE-IT study [52,54]. There is evidence that EPA at a sufficient dose (1800 or 2700 mg/d) suppressed the platelet aggregation function in patients on antiplatelet therapy [55]. Therefore, high doses of n-3 PUFAs, especially EPA, might have a stroke prevention effect. Currently, the Statin Residual Risk Reduction with Epanova in High Cardiovascular Risk Patients with Hypertriglyceridemia (STRENGTH) trial is being carried out to assess the benefit of high doses of both EPA and DHA therapy for the prevention of cardiovascular diseases [56].

Although the JELIS and REDUCE-IT studies did not show increases in any hemorrhagic events, to investigate the hemorrhagic complication rate in the secondary prevention of stroke after combination therapy involving antiplatelet or anticoagulant agents with high-dose n-3 PUFAs is of importance. Moreover, recent evidence highlights that protein convertase subtilisin/kexin type 9 inhibitors and monoclonal antibody targeting interleukin-1β are effective for lipid management [57,58]. Thus, therapy for abnormal lipid metabolism can be diverse, and the efficacy and safety of n-3 PUFAs for the secondary prevention of stroke in real-world clinical practice, sometimes with such novel agents, are yet to be elucidated.

## 4. Levels of n-3 PUFAs as Surrogate Markers of Stroke

The n-3 PUFAs can be measured in adipose tissue, erythrocytes and plasma, and there has been growing interest in plasma levels of n-3 PUFAs as potential critical surrogate markers for stroke in population- and hospital-based studies. So far, many studies have examined plasma PUFA levels, with a significant number focusing on the adipose tissue and erythrocyte content of PUFAs [31,59,60,61,62].

In terms of the erythrocyte membrane content of PUFAs, the Framingham Heart Study, using a population of 2500 people, showed that there was an inverse association between n-3 PUFA index and stroke incidence and a clear relationship between n-6 to n-3 PUFA ratio and stroke incidence. However, EPA and DHA levels did not correlate with stroke incidence [63]. In three large cohorts in the USA, including the Cardiovascular Health Study (CHS), Nurse’s Health Study (NHS), and Health Professional Follow-up Study (HPFS), DHA was inversely associated with atherothrombotic infarction (HR, 0.53; 95% CI, 0.34–0.83), and docosapentaenoic acid (DPA) was inversely related to cardioembolic stroke (HR, 0.59; 95% CI, 0.38–0.93), while there was no significant association between EPA and the incidence of any stroke subtype [64]. In this study, PUFAs in the erythrocyte membrane were examined in the NHS and HPFS [64]. Another study in Korea showed that n-3 PUFA index was linked with both hemorrhagic and ischemic stroke [60]. On the other hand, a large-scale cohort study in Denmark that analyzed levels of PUFA in adipose tissue demonstrated that the highest quartile of EPA was associated with low incidences of all-cause stroke (HR, 0.74; 95% CI, 0.62–0.88), large artery atherosclerosis (HR, 0.52; 95% CI, 0.36–0.76), small vessel occlusion (HR, 0.69; 95% CI, 0.55–0.88), and stroke due to other causes (HR, 0.52; 95% CI, 0.27–0.98) [31].

In terms of plasma levels of PUFAs, we previously showed in 373 patients with acute ischemic stroke that EPA/AA and DHA/AA ratios were significantly lower in patients aged <65 years than in patients aged 65–74 and ≥75 years, together with a higher frequency of current smoking, higher body mass index and higher serum triglycerides. Additionally, ischemic stroke patients aged <65 years had lower EPA/AA and DHA/AA ratios than healthy subjects of the same age [62]. Our data indicated that lower EPA/AA and DHA/AA ratios might contribute to the development of ischemic stroke in younger patients, in association with lifestyle risk factors [62]. A previous meta-analysis showed that plasma n-3 PUFA levels were inversely associated with stroke incidence (RR, 0.86; 95% CI, 0.76–0.98), especially levels of DPA (RR, 0.74; 95% CI, 0.60–0.92) and DHA (RR, 0.78; 95% CI, 0.65–0.94), but not EPA [59]. There was a prominent association between n-3 PUFA levels and ischemic stroke (RR, 0.81; 95% CI, 0.68–0.96), but not with hemorrhagic stroke (RR, 0.95; 95% CI, 0.60–1.49) [59]. In a study within 24 h of the onset of acute ischemic stroke in 281 patients, EPA/AA, DHA/AA, and EPA+DHA/AA ratios showed a negative association with neurological deterioration and a ≥2-point increase in the National Institute of Health Stroke Scale (NIHSS) score within a 72-h period [61]. Another study demonstrated that lower proportions of EPA, DHA and Σω3-PUFAs were linked with stroke severity on the NIHSS score, and in particular, that DHA and Σω3-PUFAs predicted poor functional outcomes (modified Rankin Scale score ≥3) at 3 months [65].

The aforementioned data indicated that different methods of n-3 PUFA analysis produce different results, and moreover, that each method of analysis leads to diverse results. Generally, plasma levels of PUFAs reflect recent dietary intake, while erythrocyte and adipose tissue content of PUFAs reflect long-term dietary consumption [66]. Further studies are needed to confirm these observations.

## 5. Biological Effects of Omega-3 PUFAs for Stroke

### 5.1. Omega-3 PUFAs and Their Metabolites

EPA competitively inhibits prostaglandin E2 formation by cyclooxygenase (COX) 1/2 from AA, and it produces less-inflammatory prostaglandin E3, thereby showing an anti-inflammatory effect, inhibition of monocyte adhesion and platelet aggregation, and improvement of endothelial injury [67,68]. EPA also decreases the production of mediators and enzymes from inflammatory cells such as macrophages and stabilizes atherosclerotic plaques [22,23,24]. There is evidence that DHA suppresses inflammation more potently than EPA [69,70]. Importantly, EPA and DHA are metabolized via the COX and lipoxygenase pathways into a new class of lipid mediators [67,68]. Specialized pro-resolving lipid mediators, including resolvins, maresins, and protectins, are synthesized from n-3 PUFAs. In the metabolism of EPA, 18-hydroxyeicosapentaenoic acid (18-HEPE) is converted from EPA in the cyclooxygenase-2 or cytochrome P450 pathways, and further metabolized to the resolvin E series including resolvin E1, E2, and E3 (Figure 2) [71,72]. In the metabolism of DHA, 12-lipoxygenase converted DHA to maresins, while the resolvin D-series were converted from DHA to the intermediate 17S-hydroperoxy-DHA by 15-lipoxygenase and further metabolized by 5-lipoxygenase. Protectins were converted from DHA by 15-lipoxygenase. These metabolites have powerful anti-inflammatory effects. In experimental stroke models, it has been shown that maresins, protectins, and resolvin D-series derived from DHA exerted an anti-inflammatory response, ameliorated stroke injuries, and induced neurogenesis and angiogenesis [73,74,75,76].

### 5.2. Treatment with Omega-3 PUFAs in Experimental Stroke

So far, underlying molecular mechanisms after supplementation with n-3 PUFAs have been studied in vivo and in vitro. In a middle cerebral artery occlusion (MCAO) model in rodents, it was shown that treatment with n-3 PUFAs prior to MCAO [77,78,79,80], after MCAO [81,82], and both [83] decreased infarct volume and improved neurological deficits and motor function. In vitro, n-3 PUFAs suppressed lipopolysaccharide-induced nitric oxide and tumor necrosis factor-α release and the inflammatory response, altered the shift from M1 to M2 phenotype, enhanced myelin phagocytosis in cultured microglia, and activated nuclear factor E2-related factor 2 and upregulated heme oxygenase-1 in neurons [82,84,85]. With regard to neuroprotective effects of EPA, suppression of oxidative stress and endothelial Rho-kinase activation were induced after ischemia [79]. In ovariectomized rats subjected to transient MCAO, EPA regulated post-ischemic high mobility group box 1/toll-like receptor 9 pathway proliferator-activated receptor gamma-dependently and -independently [80]. On the other hand, DHA reduced the expansion of infarct area due to a subsequent inflammatory response elicited by ischemia, via the promotion of conversion from 15-lipoxygenase-1 to neuroprotectin D1, exhibiting cell-protective, anti-inflammatory, and anti-apoptotic responses [86]. Other studies showed that DHA induced neurological recovery and reduced infarct size by diminishing blood-brain-barrier damage, regulating microglial infiltration, and reducing oxidative stress and activating AKT cascades in neurons [85,87,88] (Figure 2).

### 5.3. Role of Omega-3 PUFAs and Their Metabolites before and after Stroke

The aforementioned data indicated that various anti-inflammatory effects, anti-oxidative stress, diminishing blood–brain-barrier damage, and regulation of signaling pathways have been implicated in the mechanisms of the roles of n-3 PUFAs against brain injury after stroke, which were clearly shown in vivo and in vitro in rodents. It is reasonable that treatment with PUFAs prior to stroke displayed a protective effect against ischemic injury, which might be consistent with the results from previous epidemiological and other studies, showing partial effects for reduction of stroke burden. Likewise, a beneficial effect of n-3 PUFA treatment immediately after stroke could indicate that the n-3 PUFAs are potential candidates for acute stroke therapy (Figure 3). In clinical trials involving stroke patients, no evidence has been available on the effects of n-3 PUFA supplementation for stroke outcome. Future research may facilitate the development of possible treatment agents for ischemic stroke in humans.

## 6. Conclusions

Since a series of epidemiological studies of the Greenland Eskimo population in the late 1970s, numerous epidemiological studies and large-scale clinical trials have been carried out. However, the effectiveness of n-3 PUFAs for secondary prevention of stroke is yet to be elucidated. Further study is warranted to show that dietary intake of or treatment with n-3 PUFAs reduces stroke burden and mortality, which may offer a potential candidate for public health benefit to prevent stroke, as well as cardiovascular diseases.

## Figures and Tables

**Figure 1 ijms-20-05549-f001:**
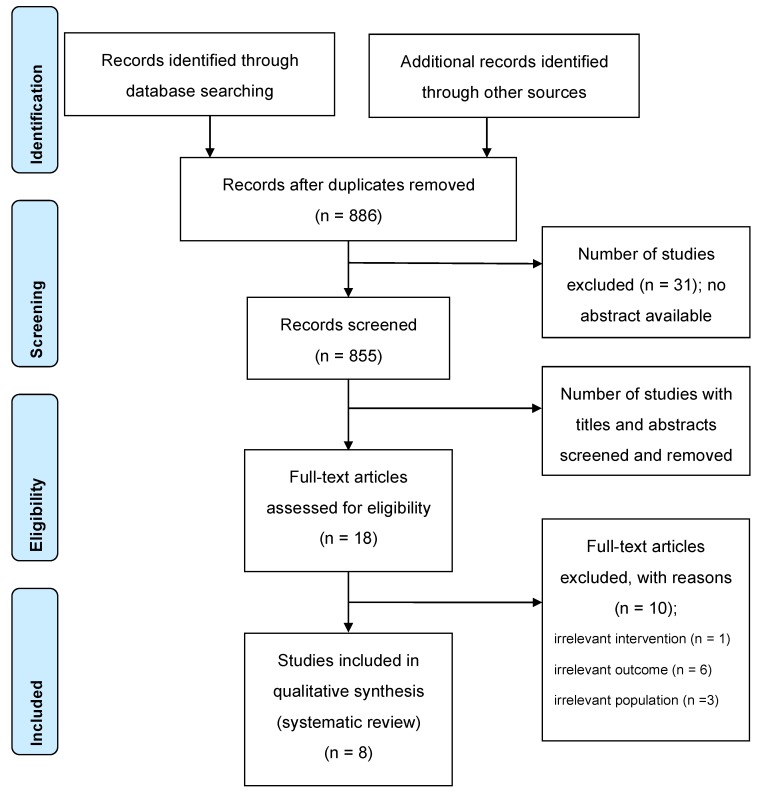
Flow diagram of the systematic literature search. Preferred Reporting Items for Systematic Reviews and Meta-Analyses (PRISMA) flow diagram indicates the number of records identified, included, and excluded through the different phases of a systematic review.

**Figure 2 ijms-20-05549-f002:**
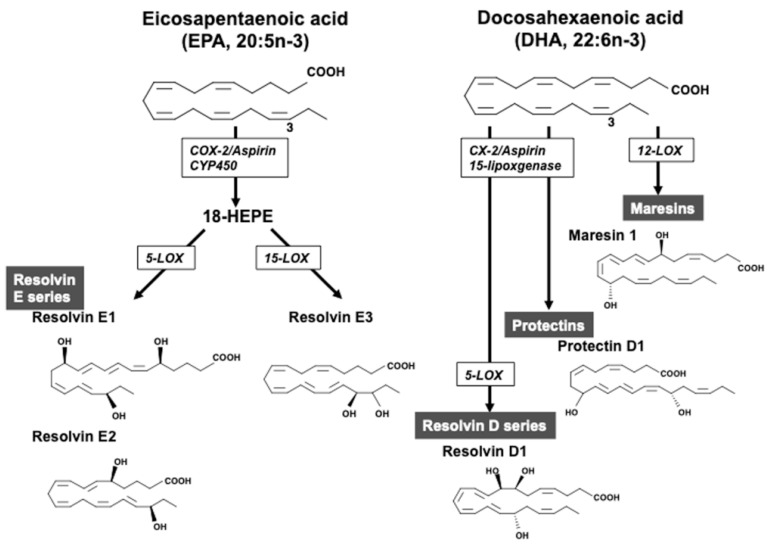
Omega-3 polyunsaturated fatty acids and specific pre-resolving lipid mediators. Eicosapentaenoic acid (EPA) is converted to resolvin E1 and E2 by 5-lipoxgenase (LOX) and resolvin E3 by 15-LOX. Docosahexaenoic acid (DHA) is converted to resolvin D1 by cyclooxygenase (COX)-2/aspirin/15-LOX, and further 5-LOX, protectin D1 by COX-2/aspirin/15-LOX, and maresin 1 by 12-LOX.

**Figure 3 ijms-20-05549-f003:**
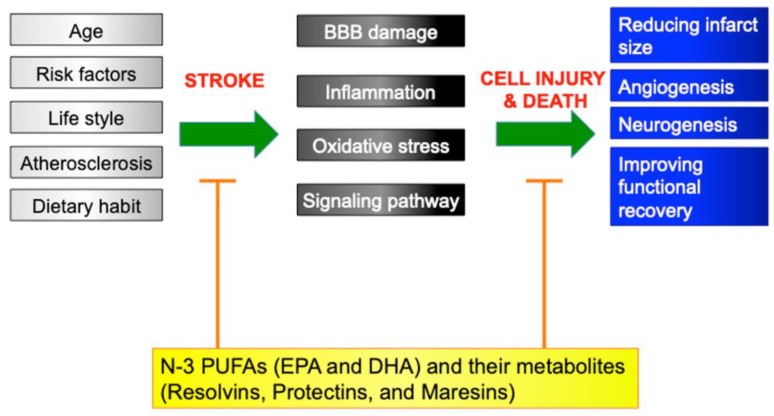
Scheme of underlying mechanisms before and after stroke. Age, cardiovascular risk factors, atherosclerosis, lifestyle, and dietary habits could be implicated in the mechanisms of stroke development in humans. On the other hand, previous experimental studies showed that the blood–brain barrier (BBB), inflammation, oxidative stress, and pathologic signaling pathways contributed to the mechanisms after stroke. N-3 polyunsaturated fatty acids including eicosapentaenoic acid (EPA) and docosahexaenoic acid (DHA), and its metabolites, resolvins, protectins, and maresins, suppressed these pathomechanisms, and reduced infarct size, induced angiogenesis and neurogenesis, and improved functional recovery. Bar-headed lines indicate an inhibition, and arrows represent a production and induction.

**Table 1 ijms-20-05549-t001:** Review of large-scale clinical trials of omega-3 polyunsaturated fatty acids.

Study	GISSI-P	JELIS	GISSI-HF	ORIGIN	SU.FOL.OM3	ASCEND	VITAL	REDUCE-IT
Subjects	Post MI (< 3 mo)	T-C ≥6.5 mmol/L1, or LDL-C ≥4.4 mmol/L	Chronic heart failure (NYHA class II–IV)	≥50 y, diabetes or prediabetes, and high risk of CVD	MI, unstable angina, and ischemic stroke	≥40 y, with diabetes mellitus, and without CVD	Men ≥50 y, women ≥55 y	≥45 y with established CVD or ≥50 y with diabetes and ≥1 additional RF, and with fasting TG level of 1.69 to 5.63 mmol/L, LDL-C, 1.06 to 2.59 mmol/L, and had been treated with statins for ≥4 weeks
Components of n-3 agents	EPA/DHA	EPA	EPA/DHA	EPA/DHA	EPA/DHA	EPA/DHA	EPA/DHA	EPA
Dosage (g/day)	1	1.8	1	1	0.6	1	1	4
No. of subjects	11,324	18,645	6975	12,536	2501	15,480	25,871	8179
Enrollment period	1993–1995	1996–1999	2002–2005	2003–2005	2003–2007	2005–2011	2011–2014	2011–2016
Follow-up (median, y)	3.5	4.6	3.9	6.2	4.7	7.4	5.3	4.9
Primary endpoints	All-cause death, non-fatal MI, and non-fatal stroke	Major coronary event(b)	All-cause death	Cardiovascular death	Non-fatal MI, ischemic stroke, and	Non-fatal MI, stroke, TIA, and vascular death excluding ICH	Major cardiovascular events(e)	Cardiovascular death, non-fatal MI, nonfatal stroke, coronary revascularization, and unstable angina
n-3 PUFA treatment results for PE, HR or RR (95% CI)	0.85 (0.74–0.98) (a)	0.81 (0.69–0.95)	0.91 (0.833–0.998)	0.98 (0.87–1.10)	1.08 (0.79–1.47)	0.97 (0.87–1.08)	0.92 (0.80–1.06)	0.75 (0.68–0.83)
Stroke outcome	Fatal and non-fatal stroke	Fatal and non-fatal stroke	Fatal and non-fatal stroke	Fatal and non-fatal stroke	Fatal and non-fatal stroke, and death from cardiovascular diseases	Non-fatal ischemic stroke	Fatal and non-fatal stroke	Fatal and non-fatal stroke
n–3 PUFA treatment results for SO, HR or RR (95% CI)	0.95 (0.61–1.47) (a)	1.08 (0.95–1.72) (c)	1.16 (0.89–1.51)	0.92 (0.79–1.08)	1.04 (0.62–1.75)	1.01 (0.84–1.22)	1.04 (0.83–1.31)	0.72 (0.55–0.93)
		0.8 (0.64–0.997) (d)						
Reference number	[44]	[51,52]	[45,46]	[48]	[47]	[50]	[49]	[53,54]

MI = myocardial infarction, mo = month, T-C = total cholesterol, LDL-C = low-density lipoprotein cholesterol, y = years, NYHA = New York Heart Association, CVD = cardiovascular disease, RF = risk factors, TG = triglyceride, EPA = eicosapentaenoic acid, DHA = docosahexaenoic acid, TIA = transient ischemic attack, ICH = intracranial hemorrhage, PUFAs = polyunsaturated fatty acids, PE = primary endpoint, HR = hazard ratio, RR = risk ratio, CI = confidence interval, SO = stroke outcome. (**a**) = calculated by four-way analysis. (**b**) = sudden cardiac death, fatal and non-fatal myocardial infarction, and other non-fatal events including unstable angina pectoris, angioplasty, stenting, or coronary artery bypass grafting. (**c**) = subgroup of primary prevention for stroke (n = 17,703). (**d**) = subgroup of secondary prevention for stroke (n = 942). (**e**) = myocardial infarction, stroke, and death from cardiovascular causes.

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
