# Peer review of "Omega-3 Polyunsaturated Fatty Acids and Stroke Burden"

_ijms, 2019, doi:10.3390/ijms20225549_

Round 1

Reviewer 1 Report

I think that the authors adequately addressed all the concerns
In my opinion, this paper is ready to be published.

Author Response

We deeply appreciate the reviewer for reviewing our manuscript. We attached Table 1 in the end of revised manuscript. 

Reviewer 2 Report

Much improved, and clearer now. There is no section on blood levels of EPA&DHA and stroke, providing rather clear data, specifically if levels were measured in erythrocytes (e.g. PMID's 29559306, 19963154,26060804, and others). This should be amended.

Author Response

We appreciate the reviewer’s comment. The acttachment is the point-to-point response to the reviewer's comments . Accordingly, we created a new section entitled ‘4. Levels of n-3 PUFAs as surrogate markers of stroke’. We also referred previous papers (Nos. 69-66) including PMID's 29559306, 19963154, and 26060804, and discussed PUFA levels by dividing the data into measurements in adipose tissue, erythrocytes and plasma (Line 280, page 7). Accordingly, we modified the sentences on line 93-98, page 3. 

We also attached Table 1 in the end  of revised manuscript. 

This manuscript is a resubmission of an earlier submission. The following is a list of the peer review reports and author responses from that submission.

Round 1

Reviewer 1 Report

The review by Ueno et al. summarizes recent literature describing the effect of n-3 poly-saturated fatty acids on stroke recovery. The review is timely and will be of interest to researchers and clinicians in different fields. The manuscript is well-written and easy to follow. It includes a review of old as well as new studies.  

Minor comments. 

In the introduction in the text describing figure 1, the authors need to add a paragraph describing resolvins, protectins, and maresins. Just mentioning these names in the figure legends is confusing. 

Figure one

In figure 1, in my opinion, photographs are not informative. What do they represent? If these represent stroke, would it be easier to depict it defying the terms?

Throughout the manuscript, the authors report a lot of results from different studies. This is a great summary, but what does it all mean? I would encourage authors to put their opinion on the matter. It can be a couple of sentences at the end of each section where the authors would express their view on the subject. 

Also, discussing the dietary intake of fish one may what to take into consideration other dietary restriction/habits such as Atkinson, keto, vegetarian, and vegan diets. 

If somebody does not eat fish, what do they eat? Also, different fish contains a different amount of n-3 PUFAs. Was it ever a consideration in any of the above studies. 

Without discrediting any studies, what, in the authors' opinion, is the source of the controversy around n-effect 3 PUFA treatments/supplementations. What studies controlled for the plasma concentration of EPA and DHA?

The biological mechanisms section is underdeveloped and needs more structure. A diagram may help to streamline this section. 

Reviewer 2 Report

Ueno et al report on a systematic review of the literature on the epidemiology of omega-3 fatty acids and stroke, intervention trials with omega-3 fatty acids on stroke, and the relationship between omega-3 plasma levels and stroke. Moreover, an attempt was made to review pertinent mechanisms. The authors present some data on the relation of dietary intake, as assessed by food questionnaires, and stroke. They also present some publications reported in some intervention trials with EPA or EPA&DHA, on cerebrovascular disease and on pertinent mechanisms. The authors conclude: “Further study is warranted to show that dietary intake of or treatment with n-3 PUFAs ultimately improves plasma n-3 PUFAs and, thereby, reduces stroke burden and mortality, which may offer a potential candidate for public health benefit to prevent stroke, as well as cardiovascular diseases.”

Introduction is lengthy and covers far too many topics. Introduction should be shortened, tightened, and used to develop the scientific question in a focused manner. There is no Methods section. Results of the review are presented in text and tables. There is no Discussion section.

Major Points:

All in all, this is an unfocused and nontransparent manuscript that does not conform to scientific standards in many ways.

The language in the entire manuscript needs to conform scientific language, as outlined in a pertinent editorial (PMID 23281431). E.g., the authors report on “effects of dietary intake of n-3 PUFAs in the epidemiological studies”, which is inadequate, since, in epidemiologic studies, there can only be correlations. Also, language needs editing to become proper English.

This is claimed to be a systematic review. However, the method of the review of the literature is not reported. It remains unclear, how and why the authors selected the publications reviewed. This is no trifle matter. Another systematic review recently published reports on very different publications on stroke intervention trials (PMID 31164089). Other important aspects also remain unclear, like why plasma levels were reviewed, why there is a section on cerebrovascular disease, or why some publications on mechanisms were selected, and not others, asf.  Why and how the publications for these sections were selected remains the authors’ secret.

The validity of memory-based nutrition data has been questioned, and with good reason (PMID 26071068). Yet, the authors present the respective data uncritically.

In no trial reported by the authors, stroke was a primary endpoint. Therefore, neither selection of participants, nor trial protocol, nor trial size were adequate to answer this question. These limitations should at least be discussed.

It is unclear, why the authors chose to review reports on plasma levels of omega-3 fatty acids. Plasma fatty acid levels have a high biological variability, and, therefore, provide very “noisy” data. Other fatty acid compartments, like erythrocytes, have a far lower biological variability. The issue of biological variability is not improved by reporting on ratios of specific fatty acids, as the authors do. Moreover, different methods of plasma fatty acid analysis produce different results. Nevertheless, the authors uncritically combine the data from the publications they somehow selected.

More points could easily be raised, but the quality of the ms has already been sufficiently demonstrated.